# Variations in the Intestinal Microbiota of the Chinese Soft-Shelled Turtle (*Trionyx sinensis*) between Greenhouse and Pond Aquaculture

**DOI:** 10.3390/ani13182971

**Published:** 2023-09-20

**Authors:** Naicheng Liu, Peng Zhang, Mingyang Xue, Zidong Xiao, Mengjie Zhang, Yan Meng, Yuding Fan, Junqiang Qiu, Qinghua Zhang, Yong Zhou

**Affiliations:** 1Yangtze River Fisheries Research Institute, Chinese Academy of Fishery Sciences, Wuhan 430223, China; nchengliu@163.com (N.L.); somnium_zp@163.com (P.Z.); xmy@yfi.ac.cn (M.X.); xiaohzau@163.com (Z.X.); zzmj965@163.com (M.Z.); mengyan@yfi.ac.cn (Y.M.); fanyd@yfi.ac.cn (Y.F.); 2Key Laboratory of Exploration and Utilization of Aquatic Genetic Resources, Ministry of Education, Shanghai Ocean University, Shanghai 201306, China; jqqiu@shou.edu.cn; 3National Pathogen Collection Center for Aquatic Animals, Shanghai Ocean University, Shanghai 201306, China

**Keywords:** Chinese soft-shelled turtle, greenhouse cultivation, pond cultivation, intestinal microbial diversity

## Abstract

**Simple Summary:**

An intuitive understanding of microbial diversity and abundances of Chinese soft-shelled turtle aquaculture is crucial to comprehending these ecosystems. Herein, the evolutionary characteristics of the bacterial communities in the Chinese soft-shelled turtle and its aquaculture water systems were investigated using Illumina MiSeq sequencing. The sequencing results revealed significant differences in the microflora compositions at the phylum and genus levels in both the intestine and aquaculture water of SSTs in greenhouse and pond culture environments. The results showed that the bacterial diversity and richness compositions in the intestinal tract and aquaculture water were the same. However, the relative abundances of bacterial communities varied. The results of this study are of great significance in understanding how the environment affects SST cultures. These data may provide valuable instructions for Chinese soft-shelled turtle aquaculture management.

**Abstract:**

The microbial community structure in aquaculture water plays an important role in the intestinal microbial diversity of aquatic animals. The Chinese soft-shelled turtle (SST) (*Trionyx sinensis*) is an important aquaculture species of high economic value in the Asia-Pacific region. An intuitive understanding of the microbial diversity and abundances of SST aquaculture is crucial for comprehending these ecosystems. Herein, the evolutionary characteristics of the bacterial communities in the SST and its aquaculture water systems were investigated using Illumina MiSeq sequencing. This experiment sampled nine SSTs from a pond outside a greenhouse and was repeated three times. The sequencing results revealed significant differences in the microflora composition at the phylum and genus levels in both the intestine and aquaculture water of the SSTs in the greenhouse and pond aquaculture environments. A total of 1039 genera belonging to 65 phyla were identified. At the phylum level, the relative abundances of *Chloroflexi* (24%), *Acidobacteria* (5%), and *Nitrospira* (3%) were higher in the greenhouse water than in the pond water. The relative abundances of *Bacteroidetes* (35%), *Actinobacteria* (8%), and *Cyanobacteria* (4%) were higher in the pond water than in the greenhouse water. The intestinal microorganisms in the SSTs experienced significant changes after the SSTs were transferred from a greenhouse culture to a pond culture environment for 28 days. After the SSTs were cultured in the ponds, we observed decreases in the relative abundances of *Actinobacteria* (39% to 25%), *Cyanobacteria* (24% to 0.8%), *Chlorobacteria* (9% to 3%), and *Firmicutes* (5.5% to 0.8%. However, we observed increases in the relative abundances of *Bacteroidetes* (2% to 35%) and *Acidobacteria* (0.3% to 25%). These results showed that the bacterial diversity and richness compositions in the intestinal tract and aquaculture water were the same. However, the relative abundances of bacterial communities varied. The results of this study are of great significance in understanding how the environment affects SST cultures. These data may provide valuable instructions for Chinese soft-shelled turtle aquaculture management.

## 1. Introduction

The Chinese soft-shelled turtle (SST) (*Trionyx sinensis*), recognized for its superior nutritional and economic significance, constitutes a prevalent component of commercial aquaculture within China, which extends to other Asian regions, including Japan, Vietnam, and Korea [1]. The SST has become a mainly cultured aquatic animal, especially in China, due to its delicious taste [2]. Meanwhile, studies have shown that the SST has a high nutritional value, since it is high in protein, low in fat, and rich in minerals [3]. Nonetheless, recurrent incidences of disease have recently posed a severe threat to the sustainable growth and health of the SST aquaculture sector [4,5]. While wild capture fisheries have remained stagnant over the past few decades, a steady expansion of aquaculture has been observed. The SST occupies a vital position in the aquaculture industry of China [6]. Currently, SSTs are commercially farmed in ponds, greenhouses, and recirculating aquaculture systems (RASs) using fresh water. These farming methods have been well-established. However, the rapid expansion of aquaculture has led to a high incidence of disease outbreaks in recent years, reducing SST production [7]. In aquaculture ecosystems, bacterial communities hold a pivotal role in functions including nutrient assimilation, water quality regulation, defense against pathogens, antibiotic resistance, and the maintenance of host health [8,9]. The long-term excessive use of feed and antibiotics in the breeding stage of the SST exacerbates water pollution and damages the water environment. Moreover, it also affects the microbial structure of aquaculture water and sediments [10]. Therefore, to protect consumer health, it is crucial to ensure the safety of these aquatic products.

The intestinal microbiota play an important role in host health and nutrient metabolism [11]. The balance in intestinal microbiota composition is the key to maintaining intestinal function and systemic homeostasis. The association between the equilibrium of the intestinal microbiota and overall health represents a current focal point of scientific inquiry. Notably, environmental influences markedly impact the structure and composition of the intestinal microbiome [12,13]. Functioning akin to an endogenous endocrine organ, the intestinal microbiota synthesize and regulate numerous compounds. These substances interact with host physiology, exerting influence on a myriad of functions at both the local and systemic levels [14]. Therefore, the intestinal microbiota are crucial for the development and healthy cultivation of the SST, and it is necessary to systematically compare the characteristics of microbial communities in different environments. There are various problems in the process of SST breeding. One of the most important problems is that SSTs will die due to the change in environment if they undergo the process of transferring from greenhouse breeding to outer pond breeding. This study aimed to compare the effects of bacterial communities on the intestinal and aquaculture water of SSTs in greenhouse and pond aquaculture environments. Based on the reasons above, the flora of the aquaculture water in the greenhouse, the pond, and the intestinal microbiota of the SSTs was investigated using high-throughput sequencing technology. This technology, which is characterized by the ability to provide comprehensive information and obtain high-resolution results, has been widely used in microbiome studies of various animal species [11]. By comparing the numbers and species found in the two environments, we could understand the impact of the environment on the aquaculture of SSTs, providing scientific support for the prevention and control technology of SST diseases. 

## 2. Materials and Methods

### 2.1. Experimental Materials

The SSTs (weight 450 ± 50 g) were collected from a turtle farm in Hubei province, China. Three greenhouses and ponds were randomly selected from the turtle farm, and nine SSTs were selected from each greenhouse pond. At the same time, nine sampling points were randomly established within the aquaculture water from the greenhouse. After sampling, the same batch of greenhouse-cultured SSTs was transferred to the pond aquaculture. The SSTs were sampled on days 1, 14, and 28 after entering the pond. The types and quantity of baits fed during the breeding period remained unchanged. The sampling processing methods were the same as above. Each SST was anesthetized with the anesthetic MS-222 (2%). After a state of no heartbeat was reached, the body surface was disinfected with 75% alcohol and dissected in the biological safety cabinet, and intestinal samples were obtained. Intestinal metabolites were identified from three mix samples. The intestinal samples were transferred to 15 mL centrifuge tubes and subsequently preserved at −80 °C. The water samples collected from the greenhouse were labeled as GW, whereas the greenhouse turtle samples were labeled as GT. A total of 500 mL of water was collected from each sampling point of the water body, and the sampling method for the greenhouse and pond aquaculture was the same. The water samples collected from the pond were labeled OW1, OW2, and OW, while the turtle samples collected from the pond were labeled OT1, OT2, and OT. For the microbial community samples from the water source, sedimentation filtration was performed first. PCR amplification was conducted using Fullsin’s Pfu high-fidelity DNA polymerase, and the amplification cycles were strictly controlled to minimize the number of cycles while ensuring consistent amplification conditions for the same set of samples. Negative controls were also included to detect microbial contamination from the environment, reagents, etc. The sex ratio of males to females was 1:1, and the age of each SST was 1 year old. All animal experiments were approved by the Animal Experimental Ethical Inspection of Laboratory Animal Centre, Yangtze River Fisheries Research Institute, Chinese Academy of Fishery Sciences (Approval ID Number: YFI 2022–zhouyong–1204).

### 2.2. DNA Extraction, PCR Amplification, and MiSeq Sequencing

Bacterial genomic DNA was extracted using a bacterial DNA kit (Omega Biotek, Norcross, GA, USA). The extracted DNA was used as a template to amplify the V3–V4 high-variable region of the 16S rRNA gene using the Gene Amp 9700 system (Applied Biosystems, MA, USA). The sequences of primers used were 341F: 5′-ACTCCTACGGGAGGCAGCAG-3′ and 806R: 5′-GGACTACHVGGGTWTCTAAT-3′ [15]. Subsequently, the samples were sequenced using the Illumina MiSeq PE300 high-throughput sequencing platform (Illumina Inc, USA). Alpha diversity and beta diversity indicators were calculated using an online platform (http://www.genescloud.cn/analysisProcess) (accessed on 11 February 2023). The operational taxonomic units (OTUs) for each sample were analyzed using principal coordinate analysis (PCoA). 

First, for samples that were not assigned to OTUs, we performed this test again. Through various unsupervised sorting and clustering methods (including the Shannon diversity index, Simpson diversity index, and *t*-test), the diversity differences between the different samples and the significance of the differences were measured [15]. At the levels of species classification and composition, we could utilize principal component analysis (PCA) and hierarchical clustering methods to organize samples into different clusters based on the composition of species abundance. Moreover, we could further assess the differences in species abundance composition between different samples [8,9]. 

### 2.3. 16S rRNA Gene Analysis

Prior to analysis, initial fastq sequence files underwent a series of processing stages. Demultiplexing was initially executed, then succeeded by quality filtering via Trimmomatic trimming and eventual merging through FLASH [9]. The following benchmarks were instituted during these procedures: (i) the truncation of reads with an average quality score below 20 over a 50 bp sliding window; (ii) exact primer matching, allowing for 2 nucleotide mismatches, and the removal of reads with ambiguous bases; and (iii) the merging of sequences with overlaps exceeding 10 bp. Operational taxonomic units (OTUs) were classified with a 97% similarity threshold utilizing UPARSE (version 7.1, http://drive5.com/uparse/) (accessed on 20 February 2023). UCHIME was used to identify and discard chimeric sequences. The RDP Classifier algorithm (http://rdp.cme.msu.edu/) (accessed on 11 February 2023) was employed to analyze the taxonomy of each 16S rRNA gene sequence against the Silva (SSU123) 16S rRNA database with a confidence threshold of 70%, using information from the Silva (SSU115) 16S rRNA database [16,17].

### 2.4. Statistical Analysis

Each sample’s community abundance and sequencing data were assessed by generating rarefaction curves [16]. Alpha diversity and beta diversity were computed utilizing Mothur software (1.45.3) [18], with principal coordinate analyses (PCoAs) determined based on the OTU level [14]. R package software (ggplot and vegan 2.3.0) was used to analyze the data [18]. The Anderson–Darling test was used to test the normality, and Tukey ‘s honestly significant difference test was used for multiple comparisons [18]. All statistical analyses were conducted using the R package software (ggplot and vegan 2.3.0), with multi-group differences analyzed via a one-way ANOVA. All data are presented as means ± SD, with *p* < 0.05 denoting statistical significance.

## 3. Results

### 3.1. Characteristics of 16S rRNA Gene Sequencing

In order to characterize the microbiota discrepancies between the greenhouse and pond cultivation environments of the SST, the V3–V4 regions of the bacterial 16S rRNA gene were sequenced utilizing the Illumina MiSeq platform. After quality control was performed and chimeras were removed, 2,089,046 high-quality sequencing reads were obtained from 12 samples belonging to four groups. The number of reads per sample ranged from 70,499 to 98,767, with an average of 87,043 reads. Table 1 presents various parameters from each sample, such as read count, coverage, operational taxonomic units (OTUs), species richness estimates, and diversity indices, at a genetic distance of 3%. Ultimately, these high-quality sequences clustered into 18,981 OTUs at 97% sequence identity, with the count of phylogenetic OTUs in each library ranging from 621 to 1039. To represent each sample’s complexity, the diversity and richness indices of all samples, derived from both the greenhouse and pond cultivation environments, were calculated. Diversity was quantified via the Shannon and Simpson indices, with Shannon index values spanning from 3.868 to 8.664 and Simpson index values ranging from 0.454 to 0.998. Each sample’s richness was assessed based on the Chao1 and ACE indices, exhibiting ranges of 603.203 to 3879.180 for the Chao1 index and 422.3 to 3643.4 for the ACE index. The bacterial species richness and diversity in the samples followed this order: aquaculture water samples ranked first, followed by intestinal samples (Figure 1). The Good coverage for each sample, used to estimate sequencing completeness, exceeded 98.9% (ranging from 98.9% to 99.8%), implying that the identified sequences can represent the majority of bacteria in each sample. Figure 2 shows the relationship between the intestinal flora samples of the aquaculture water from the greenhouse, the aquaculture water from the pond, the greenhouse SSTs, and the pond SSTs.

### 3.2. Differences in the Bacterial Community Compositions

#### 3.2.1. Differences in Intestinal Microbiota between Greenhouse and Pond Aquacultures of SSTs

Figure 3 illustrates the compositions of intestinal microbiota and water in both the greenhouse and pond aquaculture environments at the phylum level, in addition to the hierarchical clustering tree at the OTU level. The relative abundances of intestinal microbial bacterial groups at the phylum and genus levels were statistically different (*p* < 0.05). Among the 65 identified phyla, 18 had an abundance of 1% in the four groups. As shown in Figure 3, the dominant phyla (relative abundance of 5% in at least one sample) in the four groups were *Proteobacteria* (5%), *Chloroflexi* (greenhouse, 9% and pond, 4%), *Actinobacteria* (greenhouse, 39% and pond, 25%), *Firmicutes* (greenhouse, 6% and pond, 0.4%), *Cyanobacteria* (greenhouse, 24% and pond, 0.8%), *Bacteroidetes* (greenhouse, 2% and pond, 35%), and *Acidobacteria* (greenhouse, 0.4% and pond, 25%). Overall, the compositions of the dominant phyla were distinctly different among the four groups (*p* < 0.01). The dominant microbial flora in the intestinal microbiota of the SST aquaculture in the greenhouse were *Actinobacteria* (25%), *Cyanobacteria* (24%), and *Firmicutes* (6%). The dominant microflora of the intestinal microorganisms in the pond aquaculture SSTs were *Bacteroidetes* and *Acidobacteria*. The relative abundances of *Actinobacteria*, *Cyanobacteria*, and *Firmicutes* in the intestines of the aquacultured SST were higher in the greenhouse than the pond aquaculture samples. The relative abundances of *Actinobacteria*, *Cyanobacteria*, and *Firmicutes* in the intestines of the SSTs aquacultured in the greenhouse were 42%, 23%, and 7%, respectively, whereas in the pond aquaculture, they were 25%, 0.8%, and 3%, respectively.

The dominant bacterial genera in the intestinal microbiota of the SSTs were *Cytophaga*, *Comamonas*, *Hydrogenophaga*, *Pseudomonas*, *Caldilinea*, *Rubrivivax*, *Polynucleobacter*, *Synechococcus*, and *Nitrospira*. The relative abundances of *Comamonas*, *Caldilinea*, and *Synechococcus* were higher in the greenhouse aquaculture than in the pond aquaculture.

#### 3.2.2. Differences in Bacterial Community Composition between Greenhouse and Pond Cultures of SSTs

Figure 4 shows the comparison of the classification and composition of bacteria at the phylum level in the aquaculture water from the greenhouse and pond cultivation. The abundances of *Bacteroidetes*, *Actinobacteria*, and *Cyanobacteria* were higher in the aquaculture water from the pond than from the greenhouse. The abundances of *Chloroflexi*, *Acidobacteria,* and *Nitrospira* were higher in the aquaculture water from the greenhouse than from the pond. The dominant bacteria at the genus level in the aquaculture water of the SSTs were *Flavobacterium*, *Caldilinea*, *Hydrogenophaga*, *Comamonas*, *Xanthomonas*, *Streptomyces*, *Saprospira*, *Myxococcus*, and *Pseudomonas*. At the genus level, the relative abundances of *Caldilinea*, *Comamonas*, *Streptomyces,* and *Myxococcus* were higher in the aquaculture water from the greenhouse than from the pond. *Flavobacterium*, *Hydrogenophaga*, *Xanthomonas,* and *Streptomyces* were higher in the aquaculture water from the pond than that from greenhouse water. *Flavobacterium*, *Aeromonas*, *Hydrogenophaga*, *Bdellovibrio, and Streptococcus* were higher in the aquaculture water from the greenhouse, which contained more opportunistic pathogen species, than that from the pond (Figure 5).

### 3.3. Environmental Impact on the Changes of Intestinal Flora in SSTs

The SSTs grown in the greenhouse for one year were transferred to the pond for aquaculture. To analyze the changes in the intestinal microbiota composition in the SSTs within 28 days, three samples were taken, on days 1, 14, and 28, after the SSTs were transported to the pond. The classification of the dominant bacteria in the intestinal microbiota of the SSTs cultured in the pond changed from *Actinobacteria*, *Cyanobacteria*, *Chloroflexi*, and *Firmicutes* to *Actinobacteria*, *Bacteroidetes*, and *Acidobacteria* (Figure 6). The relative abundance of *Actinobacteria* in the SST intestines in the pond aquaculture decreased from 39% to 30% and finally to 25%. Similarly, the relative abundance of *Cyanobacteria* decreased from 24% to 12% and finally to 0.8%, while the relative abundance of *Chloroflexi* decreased from 9% to 5% and finally to 3%. The relative abundance of *Firmicutes* also decreased, from 5.5% to 3% and finally to 0.8%. On the other hand, the relative abundance of *Bacteroidetes* increased from 2% to 20% and finally to 35%, while that of *Acidobacteria* increased from 0.3% to 16% and finally to 25% (Figure 6).

### 3.4. Functional Prediction of the Microbiota

The functional roles of the microbiota within the intestines of the SSTs raised in both the greenhouse and open-pond environments were depicted via PICRUSt (Phylogenetic Investigation of Communities by Reconstruction of Unobserved States). The putative genes were categorized, based on alignment, into eggNOG databases (KEGG Pathway Database, http://www.genome.jp/kegg/pathway.html) (accessed on 23 April 2023) (Figure 7). The functional profiles across all groups remained fairly consistent in comparison to the taxonomic profiles. Across all groups, several pathway functions were amplified, including amino acid transport and metabolism (8.21–8.54%), energy production and conversion (7.02–7.64%), cell wall/membrane/envelope biogenesis (6.19–7.06%), signal transduction mechanisms (6.30–6.99%), carbohydrate transport and metabolism (5.96–6.83%), transcription (5.34–6.97%), replication/recombination and repair (5.29–6.08%), and inorganic ion transport and metabolism (5.74–5.95%). However, the categories of “function unknown” (8.96–9.59%) and “general function prediction only” (8.64–9.02%) deviated from this trend. The COG functional classification results suggested that the intestinal microbiota of SSTs perform identical biological functions to those in sediment, regardless of whether the SSTs are cultivated in a greenhouse or a pond.

The 111 differentially expressed proteins underwent KEGG pathway enrichment analysis utilizing the online KEGG Pathway Database (http://www.genome.jp/kegg/pathway.html) (accessed on 23 April 2023). The metabolic pathways were stratified into six primary categories: Metabolism, Genetic Information Processing, Environmental Information Processing, Cellular Processes, Organismal Systems Regulation, and Human Diseases. In both the greenhouse and pond environments and in the intestinal microflora of the SSTs, the Metabolism pathways were the most enriched, followed by the Genetic Information Processing pathways. Cellular Processes and Environmental Information Processing accounted for a relatively low proportion. Almost no metabolic pathways were observed between Organismal Systems and Human Diseases (Figure 7).

## 4. Discussion

Microbial communities are an important part of aquaculture. The growth and survival rate of aquatic animals is significantly affected by the structure and function of the microbial community [19]. A better understanding of bacterial community dynamics in aquaculture systems is critical to optimizing disease management strategies [20,21]. The differences in the relative abundances of the bacteria in the aquaculture water can be attributed to various factors, such as temperature, water, light, the use of antibiotics, and other reasons, thereby resulting in disparities between the greenhouse and pond cultivation. Microorganisms do not exist in isolation but instead form complex, interacting ecological webs [22]. In this study, the interactions between taxa in the intestinal and water-bacterial networks of SSTs were mainly positive (92% and 98%, respectively), indicating that mutual benefit or cooperation may dominate the intestinal and water-bacterial networks. Conversely, negative connection (51%) dominated the bacterial network of the water body, indicating that competitive interactions are more prevalent in water bacterial networks. Exploring the relationship between aquaculture environments and SST intestinal flora is of great significance to formulating sustainable strategies for SST aquaculture. At the same time, in the comparative study of the intestinal flora diversity of SSTs cultured in ponds and greenhouses, although the functional analysis of the flora showed similar functional profiles, differences in related pathways were found. Specifically, the differences in the transport and metabolism of amino acids and the utilization of energy further revealed the differences in digestive function and absorption function between pond cultures and greenhouse cultures. A pond culture environment may be affected by more natural factors and external microorganisms than a greenhouse culture environment. This may lead to the adjustment of the digestive system of the SST in the process of adapting to the environment and food, which would in turn lead to differences in the function of the flora. We can better understand the differences between pond cultures and greenhouse cultures of SSTs and further promote the improvement of breeding efficiency and the optimization of breeding environments.

The dominant bacteria types also varied across different aquaculture waters. The most dominant phyla in the pond aquaculture water were *Proteobacteria*, *Bacteroidetes*, and *Actinobacteria* [23]. The most prevalent phyla found in the aquaculture water from the greenhouse were *Firmicutes*, *Proteobacteria*, and *Bacteroidetes*. *Firmicutes* is one of the major microbes in aquaculture water. Studies have suggested that *Firmicutes* plays a role in energy storage [24]. A high ratio of *Firmicutes* induced host obesity, since releasing signals from *Firmicutes* would prompt a host to convert more blood glucose into fat [25]. The *Proteobacteria* phylum contains many functional aerobic bacteria that possess the functional abilities to eliminate organic matter and nitrogen [26]. *Proteobacteria* play a role in the cycling of several materials, such as the carbon, nitrogen, and sulfur cycles, in aquatic ecosystems [27,28]. Moreover, *Proteobacteria* are the dominant flora of the intestinal microflora of SSTs, and they can maintain the balance of this microflora [29,30]. *Bacteroidetes* was the predominant microbe in the fecal matter of the SSTs, and it mainly participates in the regulation of fat and bile acid metabolism and the maintenance of the energy balance of the host [31,32]. *Bacteroidetes* protects hosts against pathogens [33]. *Actinobacteria* are prolific producers of secondary metabolites with diverse biological activities [34]. Therefore, *Actinobacteria* are critical in maintaining intestinal homeostasis [35]. Although there were similar dominant bacterial species in the water bodies of the greenhouse and pond aquaculture, their relative abundances were different. It is worth noting that the abundances of these dominant bacteria in the two aquaculture environments will always be different. Studies on greenhouse aquaculture water have found that some dominant bacteria are beneficial to the digestion and absorption ability and energy utilization rate of SSTs. These dominant bacteria may have better habitat and resource supplies in greenhouse environments, thereby improving the digestive function of the SSTs. In contrast, the dominant bacteria found in open pond aquaculture water have a protective effect, which helps to maintain the intestinal homeostasis and health of SSTs. The existence of these dominant bacteria may be beneficial to preventing the invasion of pathogenic microorganisms, maintaining the ecological balance of the intestine, and promoting the healthy development of the SST. By studying the dominant bacteria in water, we can better understand their effects on the digestion, intestinal homeostasis, and health of SSTs. Further research will help us better understand the mechanisms of these dominant bacteria and provide more targeted management strategies for the aquaculture industry to promote the breeding efficiency and health of SSTs.

Recent studies have shown that factors such as locality and habitat play crucial roles in microbial structure, since factors such as local flora and fauna, the photoperiod, available food, and climate conditions may affect host microbiota [36]. Studies have also shown that the intestinal microbiota diversity of the Pacific white shrimp is closely related to water salinity (freshwater vs. marine aquaculture). Aquaculture water affects the intestinal microbiota structure of this shrimp [13]. Differences have also been found regarding its ability to grow at varying temperatures [37]. Other findings have indicated that the rate of early embryonic development in green turtles decreases with decreasing oxygen concentration [38]. Due to differences in aquaculture environments, the diversity of the intestinal microbiota varied between the Chinese soft-shelled turtles in ponds and those in greenhouses. Greenhouse aquaculture and pond aquaculture have their own advantages and disadvantages. At present, young SSTs are mainly cultivated in greenhouses and transferred to ponds after reaching a certain specification to increase the yield of breeding. The microbial diversity of greenhouse and pond cultures directly affects the diversity of the intestinal flora of SSTs. Environmental factors, including culture environment, may have an important impact on intestinal microbiota. Changes in these factors may change the composition and relative abundance of intestinal microbes, thereby affecting the health of the host. In addition, these changes may also affect the homeostasis of the entirety of the intestinal microbiota. It should be emphasized that the relationship between intestinal microbiota and host health is very complex, and there are many factors that are not fully understood. Further research will contribute to a deeper understanding of the impact of the aquaculture environment on intestinal flora and provide better management and strategies for improving intestinal health in the aquaculture industry.

The turtles had the same general microbiota regardless of origin, body size, or habitat, and they rapidly adapted to changes in different habitats [39]. The growth, behavior, and physiology of the homologous turtles varied under different living habitats in a short period. Environmental changes can significantly affect the intestinal microbiota of mammalian and aquatic animals [40,41]. Alterations in the aquatic environment of crucian carp (*Carassius auratus*) have altered the relative abundance of flora related to growth and development [42,43], and these differences can be attributed to differences in living space, water quality, food composition, and abundance [44,45]. When considering the similarity of the greenhouse and pond environments, we can further explore the potential impact of the environment on changes in intestinal flora. First of all, environmental factors such as geographical location, climate, rainfall, and temperature may have impacts on ecosystems and microbial living conditions [46]. These factors may cause differences in microbial communities, which in turn affect the composition and relative abundance of intestinal flora. Secondly, diet is an important factor affecting intestinal flora [47]. Although the composition and total amount of food in the two environments were basically the same, the movement of the SSTs under the greenhouse conditions was relatively small. Less exercise may lead to relatively low digestive activity, and reduction in digestive activity may affect intestinal morphology and function. In addition, adequate food intake can promote intestinal development and enhance digestive function. However, persistent and excessive feeding may reduce appetite and digestive activity, which may lead to changes in intestinal morphology. In addition, the digestive tract has a strong adaptability to new environments. In pond-cultured SSTs, the larger total amount of water may lead to different responses of microorganisms to environmental changes, thus affecting the species and relative abundances of intestinal microorganisms.

Greenhouse cultures and pond cultures are two completely different breeding environments, and their differences may lead to changes in the intestinal flora of SSTs [48]. First of all, greenhouse farming is usually a closed environment; in contrast, pond farming is in an open, natural environment. The water characteristics of these two environments and other biotic and abiotic factors will have a direct impact on the community structure of the intestinal microbiota [49]. For example, the water in the greenhouse may be more restricted and controlled and the water quality more stable, while the water in the pond will be vulnerable to the influence of the surrounding environment, such as weather changes, plants, sediment, etc. [50,51]. These environmental differences may affect the microbial diversity and abundance in each water body, which will in turn affect the digestive tracts of the SSTs. In addition, the water flora in the aquaculture environment may also affect the intestinal flora composition of SSTs through diet and other ways [52]. In greenhouse aquaculture, water bodies are often more restricted and treated, which may lead to changes in the species and quantity of microorganisms. In contrast, open-pond aquaculture waters are more susceptible to the surrounding environment and may have more complex and diverse microbial communities. For example, in the analysis of pond-cultured SSTs, it was found that the genus of related conditional pathogenic bacteria had a higher proportion, which was also a challenge for the pond-cultured SSTs. The differences in the microbial community ingested by the SSTs in different water environments may affect the composition and diversity of the microbial communities in their digestive tracts.

## 5. Conclusions

In summary, our work revealed that the abundance and diversity of microbiota in SST intestines varied between greenhouse- and pond-cultured SSTs. Differences were observed between the intestinal flora of the SSTs and the microflora in the aquaculture water. Moreover, the microbial diversity in the aquaculture water of the SSTs affected the microbial diversity in the intestines of the SST. The relative abundance of bacteria related to digestion and nutrient absorption was higher in the SSTs cultured in the greenhouse than in the pond. Although the relative abundance of bacteria related to digestion and absorption was low in the pond aquaculture, the conditionally pathogenic bacteria species were fewer in the intestines of the SSTs. Compared to the greenhouse environment, the growth rate of SSTs was inferior in the pond environment. However, the study of only the intestinal flora of SSTs in two environments has certain limitations. At the same time, there are many effects of pond cultures of SSTs. Overall, this study provided insights into the microecological regulation required for successful SST farming under the pond and greenhouse breeding methods. The findings of this study have enhanced our understanding of the microflora characteristics in aquatic ecosystems.

## Figures and Tables

**Figure 1 animals-13-02971-f001:**
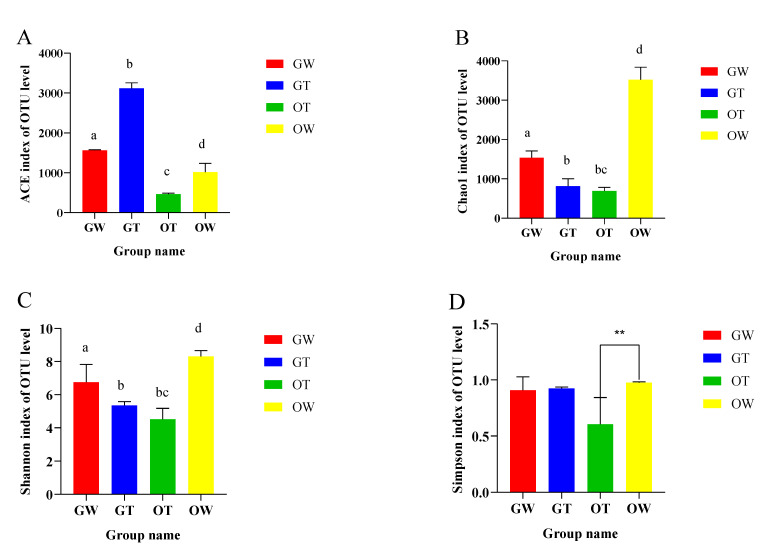
The richness and diversities of bacterial species in the four groups. The (**A**) Ace, (**B**) Chao, (**C**) Shannon, and (**D**) Simpson indices of OTU levels. ** represents *p* < 0.01. Different lowercase letters indicate significant differences between groups (*p* < 0.05).

**Figure 2 animals-13-02971-f002:**
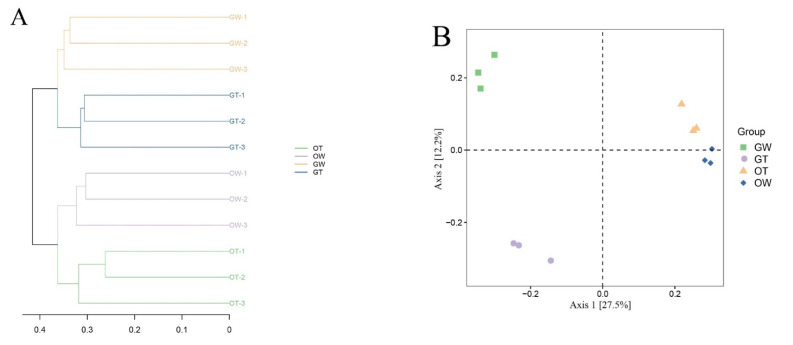
The relationships between intestinal flora samples in aquaculture water from the greenhouse, aquaculture water from the pond, greenhouse SSTs, and pond SSTs. (**A**) Hierarchical clustering tree among four samples and (**B**) principal coordinate analysis (PCoA) of the bacterial community, with three samples in each group; each sample = 3 SSTs. The hierarchical clustering tree was calculated using the UPGMA (Unweighted Pair-group Method with Arithmetic Mean) method, and the relationships between samples were determined using Bray distance and the average clustering method.

**Figure 3 animals-13-02971-f003:**
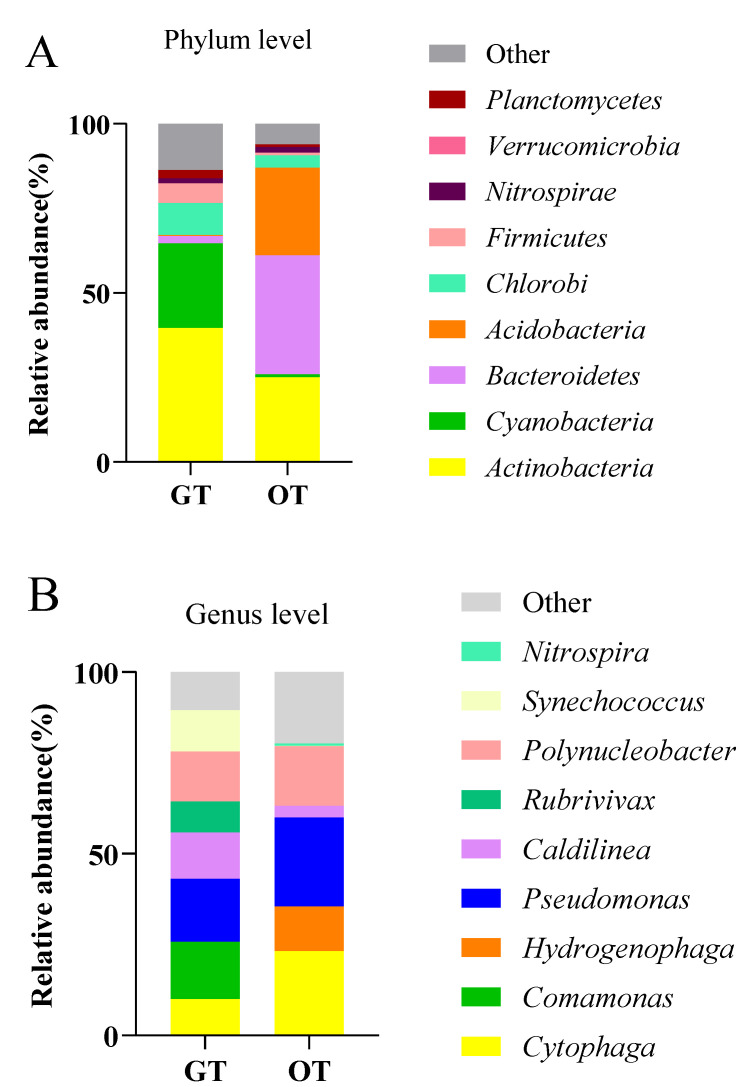
Analysis of intestinal microbes of SST aquaculture in greenhouse and pond environments. (**A**) Relative abundances of microbiota at the phylum level. (**B**) Relative abundances of microbiota at the genus level.

**Figure 4 animals-13-02971-f004:**
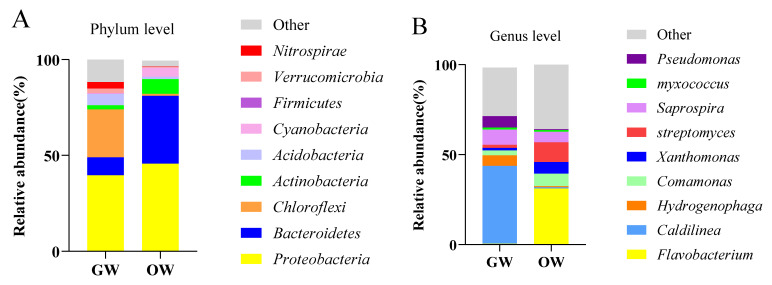
Analysis of the relative abundances of bacteria in greenhouse and pond water. (**A**) Relative abundances of bacteria at the phylum level. (**B**) Relative abundances of bacteria at the genus level.

**Figure 5 animals-13-02971-f005:**
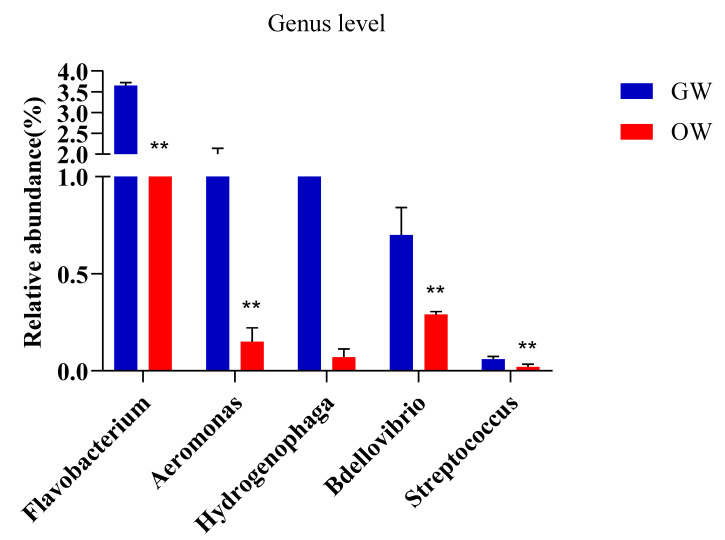
Comparison of the intestinal microbial composition of SSTs in greenhouse and pond aquaculture. The genera with more species of conditionally pathogenic bacteria were higher in the aquaculture water from the greenhouse than in the open pond. ** represents *p* < 0.01.

**Figure 6 animals-13-02971-f006:**
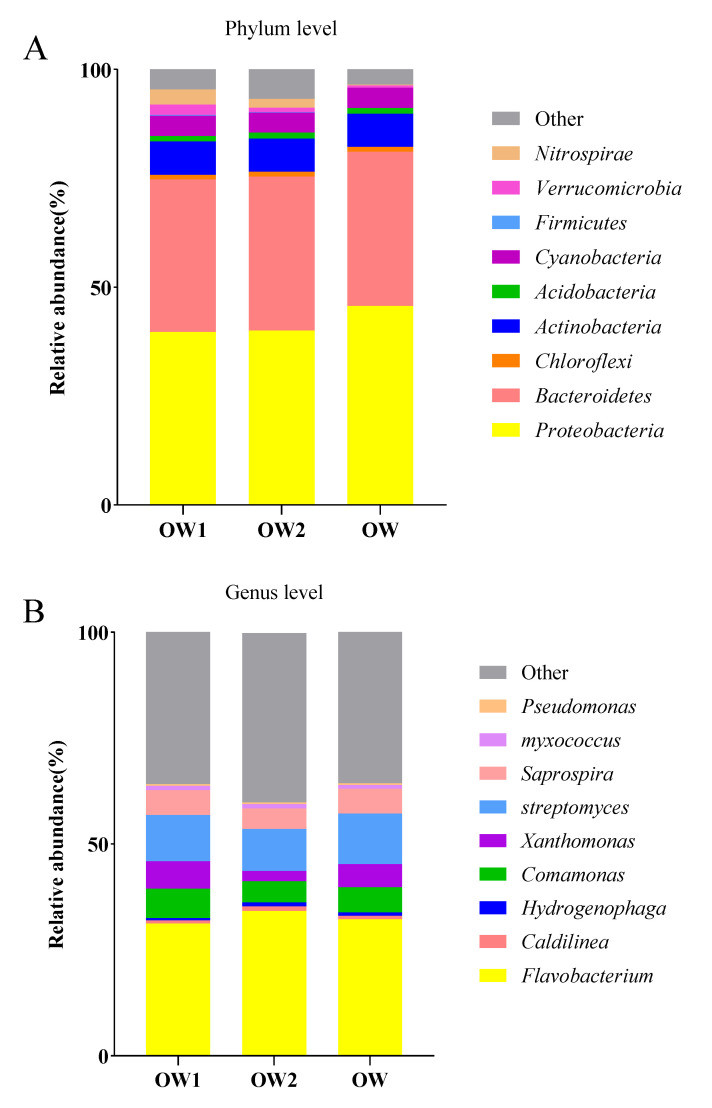
(**A**,**B**): The changes in the relative abundances of bacteria in aquaculture water from the pond within 28 days at three different times. (**C**,**D**): The changes in the relative abundances of intestinal microbiota in SSTs in the open pond within 28 days at three different times. (**A**,**C**): Relative abundances of bacteria at the phylum level. (**B**,**D**): Relative abundances of bacteria at the genus level.

**Figure 7 animals-13-02971-f007:**
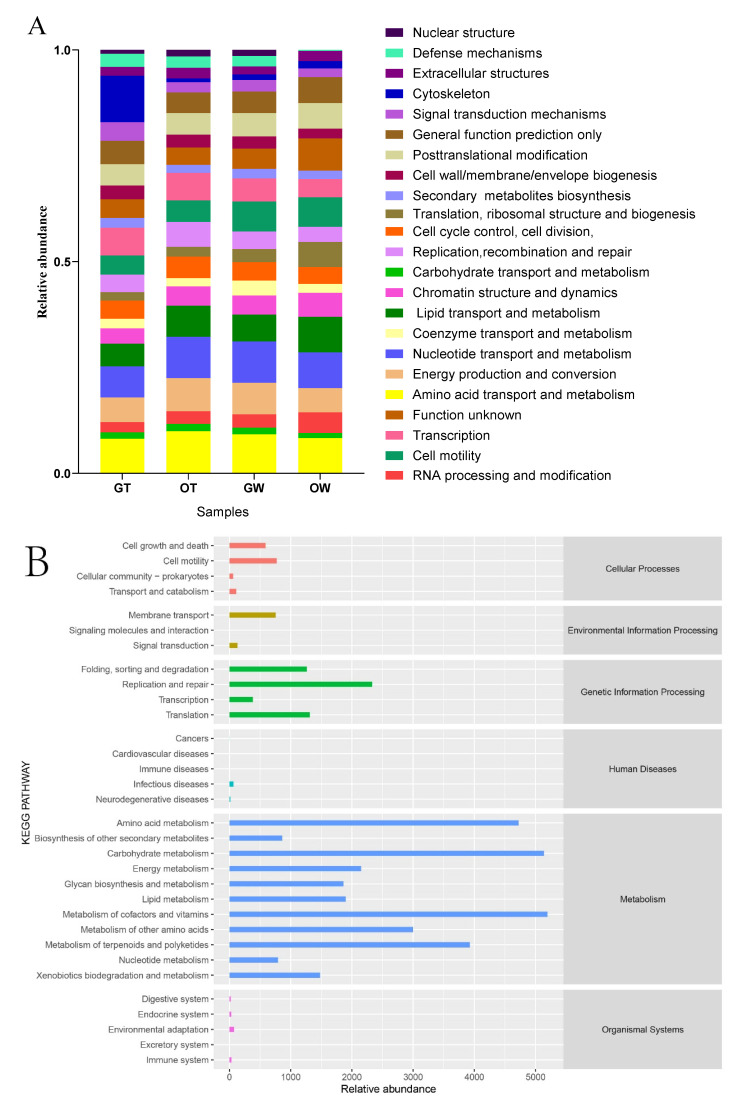
(**A**) The COG function classification of the four groups. (**B**) The KEGG pathway classification of the four groups.

**Table 1 animals-13-02971-t001:** Richness and diversity indices relative to each sample (OTU cutoff of 0.03).

Sample ID	Read Number	Coverage	Number of OTUs	Alpha Diversity			
				ACE	Chao1	Shannon	Simpson
GW-1	77,652	0.994	799	1574	1701.85	6.881	0.969
GW-2	70,499	0.993	728	1574	1359.44	7.754	0.984
GW-3	85,369	0.998	797	1551	1552.8	5.626	0.769
GT-1	83,691	0.992	869	1239.7	875.423	5.338	0.933
GT-2	85,337	0.994	887	1017.8	603.203	5.141	0.910
GT-3	89,218	0.997	1039	798.3	963.684	5.586	0.929
OW-1	97,192	0.991	649	3068.3	3879.18	8.572	0.984
OW-2	96,507	0.997	730	3273.3	3403.96	8.429	0.969
OW-3	96,507	0.993	621	3007.7	3272.43	7.911	0.972
OW1-1	93,462	0.996	667	3223.2	3434.18	8.554	0.997
OW1-2	96,757	0.994	754	3231.3	3334.76	8.664	0.965
OW1-3	98,327	0.996	665	3643.4	3232.13	7.443	0.976
OW2-1	95,462	0.994	665	3231.5	3443.38	8.334	0.955
OW2-2	98,767	0.993	768	3431.3	3654.34	8.445	0.977
OW2-3	91,347	0.993	643	3332.7	3121.12	7.944	0.966
OT-1	82,997	0.997	884	491.4	796.035	3.888	0.456
OT-2	86,878	0.995	789	465.1	664.27	4.423	0.475
OT-3	74,136	0.998	932	437.7	617.694	5.219	0.880
OT1-1	83,452	0.995	765	445.5	787.033	3.867	0.454
OT1-2	83,423	0.996	865	422.3	654.45	4.455	0.487
OT1-3	74,323	0.993	912	445.5	634.636	5.209	0.867
OT2-1	84,432	0.994	877	488.6	756.086	3.888	0.476
OT2-2	87,765	0.995	765	455.3	666.87	4.477	0.476
OT2-3	75,546	0.997	911	444.7	636.677	5.256	0.855

OTUs were defined at the 97% similarity level (threshold was 0.03). GW represents the greenhouse aquaculture water body, GT represents the intestinal tracts of greenhouse-cultured Chinese soft-shelled turtles, OW represents the aquaculture water body in the outer pond, and OT stands for the intestinal tracts of Chinese soft-shelled turtles cultured in the outer pond.

## Data Availability

All data presented in this study are available from the corresponding authors upon reasonable request.

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
