# Peer review of "Variations in the Intestinal Microbiota of the Chinese Soft-Shelled Turtle (Trionyx sinensis) between Greenhouse and Pond Aquaculture"

_animals, 2023, doi:10.3390/ani13182971_

Round 1

Reviewer 1 Report

The authors present an interesting study on microbiota of Chinese soft-shelled turtle gut (Trionyx sinensis). The bright side of the manuscript is to provide the microbiota of the species cultivated populations and is likely to contribute to the healthy production of SSTs. However, some points are missing (mentioned below) in the manuscript and some parts of the manuscript are not easy to understand. Because of these reasons, major concerns are raised. Therefore, I would like to make some suggestions to improve the quality of the paper as below:

The Introduction section needs structural changes. The Discussion section should be enriched with a more theoretical interpretation and relate the present results with additional concepts. Moreover, the limitations of the study should be given in the conclusion section.

Line 70: A few words for the usage of the high-throughput sequencing technology in microbiota studies would better fit here. Such sentence or similar sentence “The use of high-throughput sequencing technology, characterised by its ability to provide comprehensive information and achieve high resolution results, has been widely used in the study of microbiota in various animal species (references).” can be used to emphasise the importance of the high-throughput sequencing in this topic.

Lines 70-79: This part of the paper is important since the authors should explain the purpose of the study and their hypothesis (I mean; what is the problem and what did you do to solve this problem) are given here. The authors explained the purpose of the study, but I think, this part of the paper should be rephrased. Please start with a brief description of the problem, in this context importance of the detection of microbiota of SST. After, the purpose of the study should be described. Furthermore, authors may also say their contribution to aquaculture of SST with this sentence “By comparing the number and species found in the three environments, we could understand the impact of the environment on the aquaculture of SST, providing scientific support for the prevention and control technology of SST diseases”. In this way, the bridge between the problem and the study aim would be stronger.

Line 81: “2.1. Experimental materials” Please provide more information on experimental protocol.

Did the SSTs samples before the establishment?

Line 98 and Line 110: The authors should explain the methods in more detail. Specifically, how taxon assignments were performed, and how operational taxonomical units (OTUs) were detected. Also, the authors should explain in more detail how did they limit the risk of false positives. (Please see; DOI: 10.1111/aje.12817 and 10.1111/1755-0998.12749).

Lines 128-129: Please add the name of the R packages and related references to text.

Line 286: The Discussion section should be enriched with a more theoretical interpretation and relate the present results with additional concepts. For instance, the study results can be discussed with similar studies from different countries from different habitats in the broader context.

Line 350: The limitations of the study should be given in the conclusion section.

Reviewer 2 Report

Introduction

Line 42 – I recommend saying aquacultured instead of cultured here and in multiple areas of abstract and intro. Cultured has different implications so it becomes more clear if you say aquacultured instead.

Lines 70-73 – This is method and shouldn’t be in intro

Lines 75-76 – This is method and shouldn’t be in intro

Materials and Methods

Line 84-87 – Can you clarify how many samples were collected? This sounds like 9 samples from each group so 36 samples total, but in results and figures it looks like 12 total in some cases (3 per group).

Line 87-90 – You need more information on how the sample was collected from the turtles. For dissection, are they euthanized? Are you taking a tissue sample or swab? How were they stored after (RNAlater, -80)?

Line 90 – Please clarify intestinal metabolites and what the three composite samples are.

Line 91 – Gut implies all sections of the GI tract. Stick with intestine since that is what you sampled.

Line 93-94 – What was the sampling method for the water? Did you filter it? If so what was that process?

Line 100-101 – Clarify if you followed manufacturer instructions. Did you use the same method for both the water and the intestine sample?

Line 107-109 – Should be incorporated into statistical analysis section.

Line 112-113 – Describe your method for demultiplexing. Put references for Trimmomatic trimming and FLASH.

Line 126 – what alpha diversity and beta diversity metrics were used?

Line 128 – OUT should be OTU

Line 128 – explain the additional analyses.

Line 128 – What version of R did you use? Put a citation for R.

Overall statistical analysis needs some work: Did you test for normality to determine whether parametric or non-parametric tests. Explain why each test was done (i.e. to compare greenhouse water samples to pond samples, we used …). Did you adjust p value for multiple comparisons?

Results

Line 134-135 – belongs in methods

Line 137-138 – 12 samples total? Or 12 samples from each of the 4 groups? Methods say 9 samples, Table 1 shows 24 samples. There is a lot of discrepancies here that need to be rectified.

Line 178-179 – there was no mention of hierarchical clustering in your methods. You need to explain that in methods.

Line 180 – what was the p value? You need to add the test and p value in other areas too.

Line 180 – what groups were statistically different from each other? Did you do post-hoc tests to see which group in particular were different from each other?

Line 184 – distinct or significant? If significant, what was test and p value?

Line 181-196 – add percentages to all the mentions of relative abundance (like you do in abstract).

Line 206-215 – give percentages and ranges

Line 216 – how did you determine these were pathogens? Do you have a reference?

Line 248 – You need to put PICRUST in your methods with a citation.

Line 249 – You need to put citation for the eggnog databases.

Discussion

Line 295-297 – These are results

Line 312-315 – how so?

Line 332-333 – Did you test for this? There is no mention in methods or results of this.

Line 342-343 – This is your first mention of the food being different. You should add some info about the differences in the turtles’ care in the greenhouse vs pond environments in the methods.

Overall discussion – You need more. You show a lot in the results that you don’t discuss. Things like the pathogenic bacteria, PICRUSt results, etc.

Conclusion

Overall conclusion – needs to be redone. A lot of this should go in discussion. It’s not clear what you’re concluding.

Line 352-355 – Couldn’t the differences between the greenhouse SST and pond SST be due to the different amount of food offered?

Line 354-355 - Results

Line 358 – this is first mention of growth rate. Need info in your methods, results, and discussion about it.

Line 359 – what do you mean by incidence of SST?

Figures and Tables

Table 1 – You have multiple samples with the same ID number for OT

Figure 1E – not mentioned in text. Unsure the relevance.

Figure 2 – You don’t reference this figure in the text so I’m not sure where it fits in with results. You don’t mention hierarchical clustering in methods or results.

Figure 2B – number of sample points here don’t match the number of sample IDs in table 1.

Line 167 – when you say each sample = 3 SSTs, does this mean you pooled the samples? Why?

Figure 3 – Can you combine this with Figure 4 so they are all next to each and we can see the results better. Also, if you have 3 samples from GT, then you should put each individual one rather than combine them so we can see the within group variability.

Figure 5 – Caption is unclear. What is A&B referring to?

Figure 6 – In table 1 you list sample ID OW3 and OT3. In this figure you have OW and OT but no OW3 and OT3. Also in lines 96-97 you don’t mention OW3 and OT3. This needs clarification.

Reviewer 3 Report

See my comment in the attached file. Use Adobe Reader

minor errors

Reviewer 4 Report

This manuscript aimed at clarify the effects of water environment on the gut microbiota of e Chinese soft-shelled turtle (SST) through comparing the microbiota composition of the gut in SST and of the surrounding water between the greenhouse and pond culture. The experimental design made sense. The statistics was not clear to some extent. However, the presentation and the syntax of the text were poor, which made many sentences of the manuscript were hard to follow. Before publication, at least the following issues should be addressed.

Major concerns

(1) Except the data of Fig. 1A~D, no statistical values were presented in the text. It was not clear that the changes in the diversity and abundance of microbiota were significant or not. In Fig. 5, even no standard deviation was presented at the genus level of microbiota data. If the data was not significant, even the abundance value of a certain bacteria decreased or increased sharply after the SST was transferred from the green house to the pond, it might not mean a lot.

(2) The Discussion part should be almost rewritten to discuss based on their own results and data. The results of the references should be served as subsidiary to confirm the standpoint of the study.  

(3) The text was hard to follow, thus the syntax should be improved with the help of a native speaker.

 Minor issues

L5 Author names: please check the spaces

L21: water of SST? It should be presented as “the surrounding water of greenhouse and pond”

L31: bacterial composition? The composition includes both the diversity and abundance.

L46: use the abbreviation “SST”

L70: water microbiota of SST??

L73-74: syntax errors?

L77: three environments???

L96-97: Please specify the label of the samples. In my opinion, the labels of OW1, OW14 and OW28 should be better.

L99: How the water samples were pretreated to extract DNA?

L146-149: please check the significant digits of the values (3 or 4 should be enough)

Table 1: data should be presented as mean±SD………

The syntax of the text should be improved with the help of a native speaker.

Round 2

Reviewer 1 Report

The authors improved the manuscript with the previous comments. However, discussion and conclusion section were not improved. The Discussion section should be enriched with a more theoretical interpretation and relate the present results with additional concepts. Moreover, the limitations of the study should be given in the conclusion section. The manuscript still needs improvements.

Lines 77-79: Need references. Please add relevant references.

Line 114: “First, the problem samples are retested and retested.” Please rephrase the sentence.

Lines 116-122: “Through a variety of unsupervised sorting and clustering methods, combined with the corresponding statistical test methods, the diversity differences and significance of differences between different samples were measured. At the level of species taxonomic composition, the differences in species abundance composition among different samples were further measured by various unsupervised and supervised sorting, clustering and modeling methods, combined with corresponding statistical test methods.” Please clearly describe which methods were used.

Line 142: ggplxt 2 2.3.0 ? Is it ggplot? Visualization were performed using ggplot package?

Line 142: “Additional analyses were executed with the R package software (ggplxt 2 2.3.0).”  

Need a reference. Also, please explain which additional analyses were used. 

Reviewer 2 Report

Thank you for your responses to my comments. I do not believe the majority have been addressed in the manuscript. Rather than repeating here, I recommend going back and editing the manuscript to address the comments. Comments are given from a reader’s perspective so if you don’t address these in the manuscript, it will remain unclear to the reader. New comments, primarily regarding responses are below.

1.       Section 2.4 – Statistical analysis is lacking. Please include in the manuscript details on your statistical tests such as what you used to test for normality, multiple comparisons, etc.

2.       Discussion – it does not appear any edits were done to the manuscript despite recommendations from reviewers. The authors do not discuss several results including picrust results and pathogenic bacteria.

3.       Response 3: By combining the samples with such a small sample size, you are missing the variability you’d see between samples. I’d recommend looking at individual samples.

4.       Response 4: You did not answer the question. You also need to elaborate in the manuscript.

5.       Response 6, 7, 9, 12, 13, 17, 20: You need to put this information in the manuscript in a concise way.

6.       Response 16: this does not answer the question.

7.       Response 19: You gave percentages for some of them, why not all?

8.       Response 23: This does not address the question.

9.       Response 24: Please clarify whether you did or did not change the amount of food being offered between the groups of turtles (in the manuscript too).

10.   Response 26: You’ll still need to discuss why you think the different amounts of food offered are not influencing the intestinal microbiota.

Reviewer 3 Report

Dear authors, thank you for the revised version. Minor revisions are needed. As usual use Adobe Reader

Reviewer 4 Report

I have reviewed the revision of this MS. Most of the concerns have been addressed by the authors. In Table 1, the data was still presented as original ( values with too many digits) without calculating the mean and SD and without any statistical analysis. It was hard to understand the significance of the existence of Table 1.

The syntax of this MS still should be largely improved. 

Round 3

Reviewer 1 Report

The authors improved the manuscript with the previous comments. However, the manuscript still needs minor improvements as mentioned below;  

Lines 88-90: “The use of high-throughput sequencing technology, which is characterized by the ability to provide comprehensive information and obtain high-resolution results, has been widely used in microbiome studies of various animal species.”  Please add references here.

Line 132: “First, the problem samples are retested and retested.” Please phrase this sentence briefly since it is not easy to understand. Did you perform more than one test for some samples? Please explain briefly. For example, “some samples were not assigned as OTUs so we performed the test again”.

Line 132-139: “Through various unsupervised sorting and clustering methods, the diversity differences between different samples and the significance of the differences were measured (including Shannon diversity index and Simpson diversity index and t-test). At the level of species classification and composition, various unsupervised and supervised sorting, clustering and modeling methods (principal component analysis (PCA) hierarchical clustering method, used to divide samples into different clusters according to species abundance composition), and further measured the differences in species abundance composition between different samples” Please add references here.”

Lines 158-161: “R package software (ggplot 2.3.0) as used to analyze the data [18] Anderson-Darling was used to test the normality, and Tukey 's Honestly Significant Difference was used for multiple comparisons. All statistical analyses were conducted using the R package software, with multi-group differences analyzed via a one-way ANOVA”. -> “Statistical analyses and visualizations were conducted using the R software, with multi-group differences analyzed via a one-way ANOVA and Anderson-Darling was used to test the normality, and Tukey 's Honestly Significant Difference was used for multiple comparisons” (references) (also add a refence, if you used a specific R package such as MASS or vegan etc.). 

Reviewer 2 Report

Thank you for your responses and edits. I have some minor edits I recommend. I also recommend changing wording to intestines/intestinal rather than gut since you did not sample the gut. Check spelling throughout. I put comments on what I was able to find. 

1. In title, change to 'intestinal microbiota' instead of gut since that is what you sampled

2. Line 32 - change spelling to aquaculture

3. Line 42 - remove 'of'

4. Line 51 - do not capitalize soft-shelled turtle

5. Line 102-103 - can you clarify what sampling processing you are referring to? I'm not sure what methods above is referring to. 

6. Line 123 - Put the name of the DNA kit. It looks like there are a couple DNA kits sold by this company. 

7. Line 132 - Instead of saying 'problem samples', describe what the problem was. For example, 'the samples with low yield were retested', or whatever the reason for the retesting was. 

8. Line 135-139 - this sentence needs reformatting. It does not make sense as it is. 

9. Line 143 - Put a reference for FLASH

10. Line 158 - was instead of as

11. Line 183 - range is in fractions and needs to be changed to percent to stay consistent.

12. Line 185 - 186 - this sentence does not make sense as written. Perhaps you meant 'Figure 2 shows the relationship...'

13. Line 227 - the dominant what?

14. Line 229 - italicize the genus names

15. Line 281 - SST intestines, not gut

16. Line 321 - intestines, not gut. I recommend looking back through paper and adjusting anywhere that refers to the SST gut was sampled, since it was the intestines that were sampled, not gut. 

17. Line 329-300 - needs reformatting since it is unclear as written

18. Figure 1E - the venn diagram does not seem necessary. I recommend removing. 
